# Microfluidics for High Pressure: Integration on GaAs Acoustic Biosensors with a Leakage-Free PDMS Based on Bonding Technology

**DOI:** 10.3390/mi13050755

**Published:** 2022-05-11

**Authors:** Saber Hammami, Aleksandr Oseev, Sylwester Bargiel, Rabah Zeggari, Céline Elie-Caille, Thérèse Leblois

**Affiliations:** 1FEMTO-ST Institute, CNRS UMR-6174, Université de Bourgogne Franche-Comté, 25030 Besançon, France; alexandr.oseev@gmail.com (A.O.); sylwester.bargiel@femto-st.fr (S.B.); celine.caille@femto-st.fr (C.E.-C.); 2FEMTO-Engineering, 15B Avenue des Montboucons, 25030 Besançon, France; rabah.zeggari@femto-st.fr

**Keywords:** microsystems, microfluidics, acoustic biosensor, bonding technology, PDMS- SiO_2_/GaAs bonding, leakage test

## Abstract

Microfluidics integration of acoustic biosensors is an actively developing field. Despite significant progress in “passive” microfluidic technology, integration with microacoustic devices is still in its research state. The major challenge is bonding polymers with monocrystalline piezoelectrics to seal microfluidic biosensors. In this contribution, we specifically address the challenge of microfluidics integration on gallium arsenide (GaAs) acoustic biosensors. We have developed a robust plasma-assisted bonding technology, allowing strong connections between PDMS microfluidic chip and GaAs/SiO_2_ at low temperatures (70 °C). Mechanical and fluidic performances of fabricated device were studied. The bonding surfaces were characterized by water contact angle measurement and ATR-FTIR, AFM, and SEM analysis. The bonding strength was characterized using a tensile machine and pressure/leakage tests. The study showed that the sealed chips were able to achieve a limit of high bonding strength of 2.01 MPa. The adhesion of PDMS to GaAs was significantly improved by use of SiO_2_ intermediate layer, permitting the bonded chip to withstand at least 8.5 bar of burst pressure. The developed bonding approach can be a valuable solution for microfluidics integration in several types of MEMS devices.

## 1. Introduction

Microfluidics field has emerged as a solution for the precise control and manipulation of fluids at microliter scales [1,2,3]. On-chip microfluidics integration is one of the most promising development vectors, particularly in the field of biosensors [4,5]. Microfluidics-integrated lab-on-a-chip solutions have been widely used in many applications, such as clinical diagnostics on human physiological fluids, cell biology [6], detection of tumor cells, biochemical detections, electrophoresis, biochemistry, PCR [7], DNA analysis, single-cell trapping, droplets microfluidics [8], biosensors, and more [9]. Recently introduced microfluidic biosensors have the advantages of portability, high precision, easy application, and high-throughput parallel processing [10]. Ma et al. [11] showed that electrochemical biosensing made in microfluidic channels could lower the detection limit of endotoxin with a confined space and enhance van der Waals forces. They used a confined microfluidic channel and continuous flow forced the target molecules to bind to the sensing surface for fast preconcentration to enhance sensitivity and shorten detection times. Zhang et al. [12] used microfluidic channels with biosensors for detection of Salmonella using Fe-nanocluster amplification and smart phone imaging.

Micromachining processes open up the possibility to combine microsensors and microfluidics onto a single chip. Various sensing technologies have been integrated in microfluidic systems (e.g., optical, conductive, acoustic, radio frequency, and other), making assessing assess the physical properties of bio-analytes on a chip level possible. The ability to complete an assessment directly on a chip is a distinguishing feature of lab-on-a-chip solutions compared to widely spread bio-analytical tools such as surface plasmon resonance (SPR).

The integration of biosensors with microfluidics circuits is in the core of the development of integrated bio-analytical chips. Among all existing approaches, acoustic biosensors has become an important tool to study molecular interactions at the surface. Acoustic biosensors have been widely studied in the detection of gases and biomolecules [13,14]. There are different types of acoustic biosensor approaches that were developed during the last decades. Bulk acoustic wave devices (BAWs) became one of the most successful approaches in the field.

There are several materials that can be used in the fabrication of acoustic wave sensors. GaAs has been shown to be very well suited for biosensing application [15,16,17]. GaAs is a microtechnical material that combines piezoelectric properties and the possibilities for devices integration and miniaturization. GaAs can be batch micromachined using Inductive Coupled Plasma Reactive Ion Etching as well as using low-cost wet chemical etching [18,19]. In addition to its beneficial microfabrication facilities, GaAs’s surface can be chemically functionalized with alkanethiols [20], silanes, and phosphonates [21]. The microfluidics integration of GaAs BAWs has a potential to introduce a novel sensing platform.

Microfluidics solutions on the other hand are used to be built based on materials such as silicon, glass, and PMMA. These materials are commonly used to manufacture fluidic channels, taking advantage of their good mechanical properties and easiness of surface modification to immobilize affinity tags for binding target molecules on surfaces. In this field, PDMS elastomers became attractive alternative materials for microfluidics due to low cost and their remarkable physical and chemical properties, such as wide temperature range, low stiffness, chemical inertness, biocompatibility, rapid prototyping, optical transparency, non-reactivity, and high gas permeability. These features make PDMS a potential material for various applications such as pattern transfer, as well as for fabrication of the complex microfluidics systems [22,23]. In addition, its low bonding temperature (lower than 100 °C) makes it an excellent material for bonding elastomer substrates since many elastomer substrates cannot withstand a high bonding temperature.

Reproducible bonding/sealing remains one of the highest challenges for reliable applications of microfluidic systems in biosensors. The popular bonding methods such as anodic bonding for Si/Glass microfluidic devices, thermo-compression bonding, or chemical assisted bonding [24,25,26] encounter various difficulties when applied for piezoelectric substrates. On the other hand, polymers provide alternative solutions for microfluidics packaging. Recently, several new strategies were introduced to improve microfluidics packaging for the integrated sensor solutions. According to the literature, Kersy et al. used adhesion promotor GE SS4120, while it does not improve the adhesion of PDMS to Teflon [27]. It decreases the adhesion strength between PDMS-PDMS. However, this method improves the adhesion of PDMS to silicon, glass, and aluminum, which only allows the formation of a strong bond between the substrate and an un-structured layer of PDMS. Carlos Luis et al. [28] proposed the use of narrow electrode connectors for minimizing the solution leakage in the PDMS-Au interface. Yong et al. [29] used thermo-compression and laser bonding to fabricate multi-layer glass microfluidic chips. The application of sticky elastomer was introduced for epidermal electronics [30,31,32]. Heterogeneous crosslinking of PDMS was applied to enhance adhesion of PDMS to several substrates, as seen in Jeong et al. [33]. Plasma-assisted bonding was used by Xi et al. [34] for improving bonding between PDMS-coated glass cover plate and silicon substrate.

In the current contribution, we develop the solution for microfluidics integration on GaAs biosensors. In particular, we study the PDMS-GaAs system where the challenge increases due to GaAs being inert to plasma bonding. The innovation of PDMS bonding with GaAs/SiO2 substrate is finding a method to combine the fluidic cell with GaAs in order to improve biosensors. To address the challenge, some authors have attempted to increase the adhesion between the PDMS and the substrate by using a gold layer. However, this method is not suitable for seal-patterned PDMS with micromachined GaAs. Others have proposed 3-aminopropyl)trimethoxysilane (APTES) and achieved a bonding strength of 406 kPa for PMMA/PDMS bonding [35,36]. A thermal bonding method is used to bond four-layer microfluidic chip [37]. In the other approach, PDMS is mixed with a small amount of polyethylenimine solution to prepare a sticky thin layer, which works similarly to a sticky tape to adhere on glass, PMMA, and metal by contact pressing [38]. Lastly, Anil et al. [39] developed microfluidics-integrated microscale that comprise an isoporous nanostructured membrane. The GaAs on Ge/Si substrate was first flipped with the GaAs nanopyramids side bonded to a Polydimethylsiloxane (PDMS) substrate, whereas a transparent flexible polymer film was weakly bonded by Van der Waals forces [40,41]. The SiO_2_ layer was used to increase PMMA bonding capability to PDMS in fabricating gas micro valves. Ahmad et al. showed a strong and irreversible bond of PDMS on PMMA when it is covered with SiO_2_ nano particles [42].

In this study, we developed the approach for irreversible and leakage-free plasma assisted bonding to integrate PDMS microfluidic channel on GaAs substrate. For this purpose, we combined the thin-film SiO_2_ intermediate layer on GaAs substrate with plasma O_2_ treatment and low-temperature annealing. In brief, this bonding technology is obtained in four main steps: (1) SiO_2_ deposition, (2) plasma O_2_ treatment, (3) chip alignment and bonding, and (4) annealing at low temperature 70 °C. This method is appealing for its compatibility with traditional replication methods using PDMS, and the surface structures can be retained. The characterization of PDMS and GaAs/SiO_2_ surfaces before bonding was verified by contact angle, Attenuated Total Reflectance-Fourier Transformed Infrared Spectroscopy (ATR-FTIR), AFM, and SEM analysis. Bonded chips were characterized using a tensile machine strength bonding equipment on PDMS and a leakage bench test.

## 2. Materials and Methods

### 2.1. Materials

Polydimethylsiloxane (PDMS, Sylgard 184) was obtained from Dow Corning Toray Corp. SU-8 3025 photoresist was purchased from MicroChem (Newton, MA, USA). Undoped, 3-inch in diameter, and 625 ± 25 μm thick double-side polished GaAs (100) ± 0.5° wafers (AXT, Inc., Fremont, CA, USA) were used to fabricate biointerface chips. Acetone (ACP Chemicals, Saint-Léonard, QC, Canada) was used to clean substrates. Red color dye was purchased from Shanghai Macklin Biochemical Co., Ltd. (Shanghai, China). Hydrostatic pressure was applied to generate constant flow rate in microfluidic chip. Sharp blade was used to cut PDMS for placing microfluidic chip. SEM microscope (Thermofisher APREO S Low-vacuum SEM and 30 mm^2^ SDD EDX) was used to observe the microfluidic channel in detail. AFM scans of different dimensions were recorded in order to have a representative sampling of the surface roughness of GaAs/SiO_2_. The AFM cantilever had a nominal resonance frequency of 330 kHz, a force constant of 42 N/m, a length of 125 μm, and a mean width of 30 μm.

### 2.2. Methods and Equipment

The fabrication process of the PDMS microfluidic channel is schematically illustrated in Figure 1. The SU-8 mold of 70 µm thickness was fabricated with SU-8 3050 onto a 1 mm thick silicon wafer using standard photolithography processes including spin coating, pre-baking, exposure, post-baking, and development. The SU-8 mold was then used to replicate PDMS microfluidic channel. PDMS with thickness of 3 mm was prepared by pouring the mixture of low-modulus PDMS (component ratio A:B = 1:10, Sylgard 184, Dow Corning).

Air bubbles that appeared during the mixing were removed using a vacuum desiccator, followed by baking at 80 °C for 2 h, followed by pouring onto the SU-8 mold to have a sticky layer of about 300 μm. Finally, the prepared PDMS structures were peeled off from the mold and small inlet and outlet holes were punched. Images of the SU-8 mold and the fabricated PDMS microfluidic channel are shown in Figure 1. The resulting channels have a hight of 60 μm and a width of 300 μm. The GaAS surface was covered with the SiO_2_ layer by an RF reactive magnetron sputtering MP450S machine (Plassys, France). During the deposition of the SiO_2_ layer, a plasma activation process was performed in 6 mTorr, power 350 W, and with 27% oxygen flow.

Figure 2 presents an experimental setup to measure flow rate and pressure in the microfluidic channel. Once the devices were fabricated and assembled, we connected 25 mL syringes to Tygon (Sigma-Aldrich, Lyon, France) tubing to control the volume of air pumped in and out the control channels. A hydrostatic pressure is mounted to accurately displace the syringe plunger. Additionally, we connected the inlet of the microfluidic device with Tygon tubing to a flow sensor to control the flow rate of our medium and cell sample.

Hence, the pressure difference varies from 0 to 8.5 bar. With the syringe pumps, the single-layer valves in this device can be accurately controlled without using more complicated electropneumatic systems. An LG16 (Sensirion) microfuidic control system was used to deliver fluid to the channel of the test device and to monitor the applied pressure. Images of the microfluidic channel were obtained with optical microscope (Mitutoyo FS70, Mitutoyo Corp., Kawasaki, Japan) and a camera (IDS µEye, IDS Imaging Development Systems, Obersulm, Germany) with a spatial resolution of 5.5 µm/pixel. During our experiments, a flow sensor connected to a PC is placed for the continuous recording of the flow rate and pressure flow in the microfluidic channel.

## 3. Results

### 3.1. SEM Characterization

The tested PDMS microfluidic structure about 3 mm in thickness was cut along the channel length by a sharp blade. Resultant PDMS membranes with exposed microchannel (see Figure 3A) were coated with Cr thin layer to be analyzed in the SEM microscope. As shown in Figure 3B, the shape of a 400 µm wide microchannel pattern casted in PDMS in a single millimeter scale is well preserved.

To show that the presented bonding method can preserve the channel profile as pure PDMS does, we bonded a PDMS microfluidic structure to a 650 μm thick, (100) oriented GaAs substrate, previously covered with 100 nm SiO_2_ layer. Figure 3C shows the cross-sectional profile of the 80 µm high microchannel after bonding, proving that our method of chip packaging with sandwich structure is safe. The two subtrates (GaAs and PDMS) are bonded after plasma oxygen treatment of SiO_2_ intermediate thin layer and thermal annealing.

### 3.2. Activation–Characterization of PDMS and GaAs/SiO_2_ Surfaces-Interfaces

The hydrophobicity of PDMS is associated with the organic methyl groups present in the chemical structure of PDMS. The microchannel was cut out of the mold, followed by oxygen plasma treatment to render the PDMS’s surface hydrophilic. We prepared our bonding technology by a combination of surfaces treatment and annealing (see Figure 4). Oxygen plasma treatment was demonstrated to be the most rapid process to increase the hydrophilicity of PDMS’s surface by removing hydrocarbon groups and introducing polar silanol (Si-OH) groups via oxidation. The activation process duration was 60 s. During various bonding tests, we tested bonding at 50 °C and 60 °C; it turned out that the quality of the bonding is not ensured, while for the same conditions, bonding is strong at 70 °C. According to the literature. Bonding strength increases with an increase in temperature [43]; the bonding is due to the interdiffusion of polymer chains. For this, two surfaces were bonded by bringing them into contact followed by a heat treatment at 70 °C for 1 h.

To characterize the surface modifications of PDMS replica and GaAs/SiO_2_ substrate after each step of plasma treatment, contact angles (CA) measurements were performed. Water droplets (5 µL) were deposited on the surface of each studied surface. As shown in Figure 5, the contact angle dropped from 100° to 53.8° after the oxygen plasma treatment of PDMS, and from 41.3° to 11.9° after the oxygen plasma-treated GaAs/SiO_2_. The drastic decrease in contact angles indicated that the hydrophobic surface of PDMS became hydrophilic due to the hydroxyl terminals on the plasma-activated PDMS’s surface. The surface of PDMS after plasma treatment has low surface energy due to the weak intermolecular forces between the methyl groups and the strong (Si–O) and flexible (Si–O–Si) siloxane chain [44].

The surface functional group of GaAs, silicone dioxide, and PDMS were analyzed by using ATR-FTIR in the MIR spectral region from 4000 to 500 cm^−1^ (λ = 2.5–20 μm) in order to study the effect of oxygen plasma on surface modification. The IR transmittance spectra are presented in Figure 6. The peaks between 2.950 cm^−1^ and 2.970 cm^−1^ correspond to the asymmetric Si–OH bonds of PDMS. The peaks at 1.257 cm^−1^ and 1.010 cm^−1^ are attributed to CH_3_ asymmetric deformation and Si–O–Si asymmetric deformation of PDMS respectively. A high transmittance on the PDMS substrate in a visible light domain is attained. In comparison, the transmittance of the PDMS substrate is 96% and that of the GaAs/SiO_2_ is 85%, which results in PDMS transmittance being 11% better than GaAs/SiO_2_.

To ensure the success of surface modification, ATR-FTIR measurements were conducted for six different substrates to verify reproducibility with a maximum error of 5%. Due to the wafer thickness of 650 μm, GaAs disks support only internal reflections. In particular, around 1200 cm^−1^, silicon oxide possesses a vibrational mode and, thus, reduces transparency in the so-called molecular fingerprint region. The presence of a peak at 1116 cm^−1^ corresponds to a thin layer of Si-O. In addition to the transmittance peak at 1018 cm^−1^, which relates to methyl groups, there is trace of chloroform in the silicon dioxide intermediate layer. After the treatment of the oxygen plasma, a large amount of hydroxyl groups is produced on the surfaces of both the silicon dioxide and the PDMS. The later conformal contact of the two surfaces will yield a large amount of Si-O-Si bonds, which will form strong adhesion between the two surfaces. A reorganization of the short polymeric chains supporting the creation of polar groups can be considered, resulting in an increase in temperature. One major advantage of the temperature increase is that the polymerization takes place while preserving the functionality of the monomer. Moreover, the increased cross-linking density in the PDMS directly influences the strength of covalent bonding.

### 3.3. Optimization of the Bonding Microfluidic Channel with GaAs Substrate

#### 3.3.1. Test with Plasma O_2_ and Annealing

The surface of GaAs at different fabrication steps was analyzed with AFM within a scanning area of 3 µm × 3 µm. The surface morphology of the silicon dioxide was examined using the AFM, in which the grain sizes can be clearly observed and it is noticed that the surface average roughness was 293 pm. As it can be seen from the AFM images (Figure 7A,B) the grain size is in the range of 20–58 nm. For a better reading of this paper, surface morphological analyses using AFM are reported for the average of six different scans, with a maximum error of 5%.

It was found that the average roughness Ra of SiO_2_ surface decreased from 552 pm to 441 pm (Table 1) after plasma treatment. It is seen that both the average roughness and root mean square roughness of the PDMS surface are a factor around 465% higher than those of GaAs/SiO_2_.

The obtained results indicate that smoother surfaces of GaAs/SiO_2_ can be achieved by performing the above-mentioned modification. However, it was shown by Zahid et al. [45] that the plasma O_2_ treatment of PDMS leads to a significant increase in its surface roughness, from Rq = 1441 pm in the case of fresh PDMS to Rq = 40,031 pm after plasma treatment. The results further confirmed that the PDMS layer adhered on the silicon dioxide wafer during the cast molding process.

#### 3.3.2. Bonding Strength Evaluation

The bonding strength for all substrate was investigated using destructive mechanical tensile test method on square 10 × 10 mm^2^ bonded pairs. In addition to the PDMS-SiO_2_/GaAs bonding configuration, the bonding strength of two other interesting systems, PDMS-Glass and PDMS-LiNbO_3_, was evaluated. PDMS, glass, and LiNbO_3_ substrates were cut using diamond saw dicing, whereas GaAs/SiO_2_ were cut with a cleavage method. All substrates were cleaned with acetone and isopropanol to remove dust and organic contaminants. The bonding strength between the different substrates and PDMS was determined using a mechanical tester for micro-components (Nordson DAGE 4000Plus), equipped with a 250 kg cartridge (50 kg range used). The specific horizontal setups of the tensile test and the assembled sample are shown in Figure 8A. Two steel sample holders were used to sandwich the bonded sample in between. The upper part of PDMS replica and the backside of all substrates were glued on steel sample holders using a single component instant adhesive Loctite 480 (Henkel Adhesives). The sandwiched sample and its holders were fixed horizontally in the special stud-pull fixture of the tester that eliminates side forces due to gimbal construction. During the tensile test, the motorized stage moved horizontally at a uniform speed of 10 µm/s, while load force and displacement were recorded simultaneously until failure occurred. The failure mode was determined by optical inspection of fracture surfaces (interface). The tensile strength of the tested sample was calculated by dividing the measured maximal load force at the bond failure. The bonding strength was calculated as follows:(1)$$\sigma =\frac{F}{S}$$
where *F* is the load force, and *S* is the bonding area.

All tensile tests were conducted on the samples that experienced plasma treatment and thermal annealing at 70 °C for one hour. The load force curves for all tested configurations are presented in Figure 8B-left. For all samples, the strength response exhibits conventional behavior: increases over time until breakage and then sharply decreasing. Breaking was observed at 12 s, 18 s, and 19 s for GaAs/SiO_2_, LiNbO_3_, and glass, respectively. The load force for PDMS-SiO_2_/GaAs is consistently higher when compared to other configurations with glass and LiNbO_3_ substrates, i.e., 200.75 N, 128.02 N, and 77.79 N, respectively. The use of an intermediate SiO_2_ bonding layer increased the bonding strength on GaAs, resulting in a significant increase in bonding strength: increase of 55% compared with glass substrate and of 156% compared with LiNbO_3_. The achieved bonding strength of GaAs/SiO_2_ to PDMS is ~2.01 MPa.

Figure 8B-right shows the average load force for different samples bonded by the presented process. A good reproducibility (bonding tests were carried out on five similar samples) for our measurements was observed, with a maximum error of 6% caused by the dispersive imprecision on the dimensions of the bonding area. Figure 8C shows photos of broken structures after the tensile tests. In the case of PDMS-SiO_2_/GaAs chips, failure was generally observed at the SiO_2_-PDMS bonding interface. However, in the case of Glass and LiNbO_3_ samples, failure was observed in the PDMS volume which is in agreement with the literature reports [27,36,46]. The results can be explained by the higher bonding strength between PDMS-Glass and PDMS-LiNbO_3_ than the coherence force of cured PDMS materials. In fact, when PDMS’s surface is brought into contact at elevated temperatures where the molecular chain mobility is high, adhesion occurs at the interface. The chain ends penetrate to the opposite substrate, leading to high bonding strength. Moreover, the increase in surface roughness enhances the adhesion property of substrates. The surface roughness of LiNbO_3_ was bigger than glass no matter what the plasma activation time was according the findings of Xu et al. [47]. To proceed further in our interpretation, we would like to hypothesize that the large mismatch in the coefficient of thermal expansion (CTE) should be analyzed in [47,48] between LiNbO_3_ and glass (14.4 (*x*, *y*-axis)-7.5 (*z*-axis) × 10^−6^ K^−1^ for LiNbO_3_ and 0.56 × 10^−6^ K^−1^ for glass). According to this hypothesis, there is bigger stress at PDMS-LiNbO_3_ interface due to higher mismatch in (CTE) when compared to PDMS-Glass. Therefore, PDMS-Glass resists higher load.

Since the SiO_2_ layer was also locally detached from the GaAs surface, this failure mode is partially affected by limited adhesion of SiO_2_ layer on GaAs that indicates that the bonding strength is higher than the coherence of PDMS material. The high stiffness of GaAs promotes interface failure, we would like to hypothesize that the high value of the Young’s modulus of GaAs (118 GPa) implies that the fracture takes place at the interface. The observed failure of the PDMS–SiO_2_/GaAs bond was consistently at the interface between SiO_2_, indicating a very strong bond between SiO_2_ and PDMS. SiO_2_ has been reported to be a reliable adhesive layer to bond PMMA substrates on PDMS for microfuidic applications [42].

#### 3.3.3. Leakage Tests

In order to validate a leakage-free performance of the bonded PDMS-SiO_2_/GaAs chips, the channel inlet/outlet holes were equipped with epoxy-sealed connectors and the leakage test was performed under defined flow conditions (Figure 9A). The flow rate increased from 10 µL/min to 4500 µL/min, looking for the maximum working pressure with no leakage appeared. A red dye solution was used for easier optical microscope inspection of potential leakage at the border of the channel.

In the leakage test, the design chips based on PDMS-SiO_2_/GaAs were kept intact (i.e., without dissemination of red dye solution into the bonded interface) until 8.5 bar of working pressure, which is maximum available in our experimental setup. It is clearly seen in Figure 9A that the channel is well defined, and no leakage is observed at flow rate of 4500 µL/min and 8500 mbar (the limit of control pressure sensor). The maximum working pressure and the maximum flow rate are shown in Figure 9B.

## 4. Discussion

Due to the challenges related to the use of PDMS as the structural material, most studies have focused on finding alternative materials. Although other polymers, such as PS, PMMA, TPE, COP, and photoresists, have been used for the fabrication of microfluidic devices [46,49,50,51]. Many different problems related to the bonding of the microfluidic channel on piezoelectric substrate such as GaAs (100) were identified. Bonding processes are required for assembly of microfluidic devices, made of two or more components. This can be achieved by using of double-sided tape, glues, or solvent bonding [52]. The bonding of PDMS on different substrates has been reported in the scientific literature [36], but the solvents used in the bonding process can also strongly influence the growth of cells cultured in the microfluidic devices [53]. Wu et al. used (3-Mercaptopropyl) trimethoxysilane (MPTMS), which was a chemical coupling reagent to modify the surfaces of the noble metals, and PDMS to improve their adhesion [54]. Yong et al. proposed a novel approach to fabricate multi-layer glass microfluidics chips, which comprises laser cutting and thermocompression bonding [29]. Among the methods presented above, special treatments processes or the processes of adding additional chemical reagents are required to achieve PDMS adhesion. This work presents an approach to assess how to improve PDMS/GaAs bonding based on the combination of low temperature (<100 °C) plasma/thermal treatments and the use of appropriate intermediate bonding layer. We proposed a new solution for bonding of PDMS microfluidic cell on the gallium arsenide substrate, covered with silicon dioxide thin intermediate layer. The aim of the characterization step was to optimize the bonding quality of multilayer PDMS-SiO_2_/GaAs. The study consisted of experimental investigations. We have characterized the bonding interface by various measurement techniques (SEM, AFM, ATR-FTIR, and CA).

In order to prove the usefulness of the proposed solution, the elements and the chemical bonds on PDMS’s surfaces have been determined by ATR-FTIR analysis. The roughness and topography of various treated and non-treated PDMS and GaAs/SiO_2_ surfaces were also analyzed using AFM. The PDMS-SiO_2_/GaAs samples fabricated according the proposed method were able to withstand the load force until 200.7 N without failure, which corresponds to bonding strength of 2.01 MPa, which was the highest value we obtained. This is substantially higher than the bonding strength of other tested microfluidic systems, such as PDMS-Glass or PDMS-LiNbO_3_. An analysis system was created to measure the bonding strength of the bonded chips. Zhen et al. showed a bonding strength of over 1.4 MPa for PDMS and PMMA [36]. Yong et al. reported the optimal pressure 0.4 MPa [29]. Kersey et al. employed that the adhesion promoter GE SS4120 can improve the adhesion of PDMS to silicon, glass, and aluminum substrates, with bonding strength values 0.841 MPa, 0.847 MPa, and 0.488 MPa, respectively [27]. From our experiment, the bonding strength obtained is higher than other PDMS bonding methods.

In the plasma treatment process, there is only a small amount of hydroxyl groups (the inherent hydroxyl groups) on the surfaces of PDMS and silicon dioxide layer. Xiangdong et al. mentioned that the contact of such two surfaces will produce a small amount of Si-O-Si bonds, which can only form a week adhesion force [55]. Thermal annealing is another method used to improve adhesion performance. Indeed, annealing contributes to the stabilization of the bonding layer, improves the cross-linking density in the PDMS and favors the orientation of molecular chains. Although a bonding temperature around 100 °C is low for silicon or glass substrates, it affects the bonding performance of polymer substrates greatly. Winnie et al. [56] used hot embossing techniques for bonding the PDMS-PMMA system at 90 °C for 3 h. An additional annealing process at 65 °C for 1 h on PDMS and PS surfaces improved bonding and allowed the stabilization of higher SEF for a longer period of time (>3 days). In this work, to enhance bonding, we have combined plasma treatment and annealing. In our experiment, after plasma treatment, both parts are aligned and pressed together while undergoing the curing process at low temperature of 70 °C for 1 h. Pre-stress uniformly applied during bonding significantly influences the orientation of the polymer chains. Hammami et al. [57] showed that the combination of temperature and stretch promotes the orientation of molecular chains in the dielectric elastomer. Subsequently the combination of temperature and small deformation promotes bonding of our system.

The chip holder ensures that the leakage could only occurs at the bounded interface between PDMS and GaAs/SiO_2_ substrates, i.e., not at the tubing connector. No leak was observed in the tested PDMS-SiO_2_/GaAs samples until maximal available working pressure of 8.5 bar. To our knowledge, it is the highest leakage-free pressure reported in the literature for PDMS-based bonding systems. A comparison between previously reported results of leakage test is shown in Table 2.

Overall, this study presents a method for evaluating whether PDMS can be used as a reliable material for microfluidic devices in order to enhance the performance of acoustic-fluidics-biosensors based on GaAs. Our contribution is discussed only in terms of basic technological challenge to couple PDMS with GaAs. Nevertheless, the presented results are likely to contribute to the improvement of the performance of the microfluidics systems combining PDMS and GaAs.

## 5. Conclusions

We have presented a novel bonding solution between GaAs and PDMS, which enables the development of an increasingly in-demand array in biosensors field, including those requiring high flow rates and high pressures. The combination of SiO_2_ intermediate layer with plasma oxygen and low-temperature annealing (70 °C) significantly improves the bonding of PDMS to the GaAs substrate. In our acoustic biosensor application, one can assume a maximum pressure driving around 8.5 bar for the fluid. The bonding area of microfluidic devices can withstand a stress about 2.01 MPa. Additionally, this bonding method does not require wet chemical treatments on bonded surfaces, which may be prohibited in some applications. Bonding features were evaluated using different methods, bonding strength, and leakage tests. Compared to the previous studies, our included bonding method distinguished a robust and rapid fabrication technology as well as superior bonding strength and leakage-free pressure. The obtained results can be valuable for the research and development of integrated microfluidic devices based on PDMS material in general.

## Figures and Tables

**Figure 1 micromachines-13-00755-f001:**
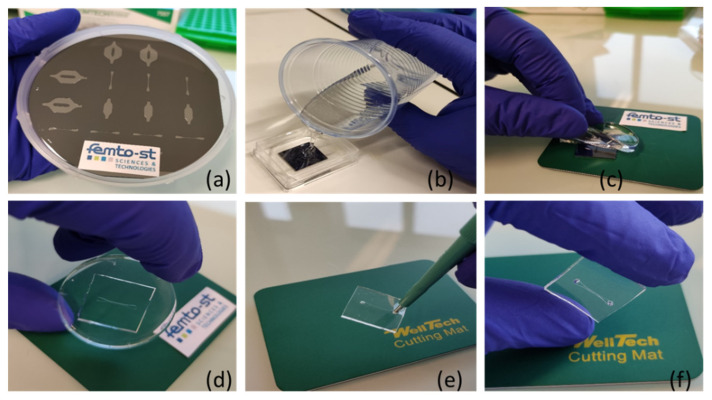
Fabrication process of the PDMS microfluidic channel by replica molding: (**a**) fabrication of SU-8 master mold using photolithography; (**b**) pouring of the mixture of PDMS prepolymer and curing agent into the master mold and allowing it to solidify; (**c**,**d**) peeling of the solidified PDMS from the master mold and cutting; (**e**) punching the inlet and outlet holes; (**f**) microfluidic channel.

**Figure 2 micromachines-13-00755-f002:**
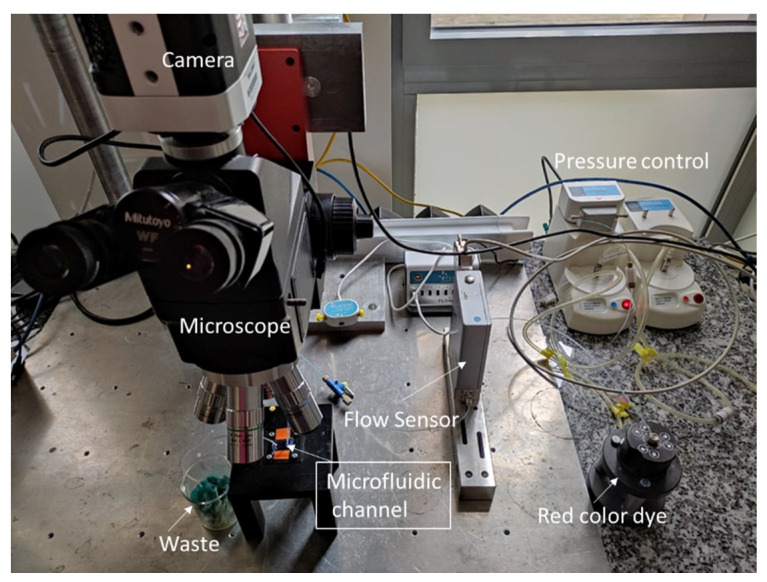
Experimental setup to control flow rate and pressure in the microfluidic channel.

**Figure 3 micromachines-13-00755-f003:**
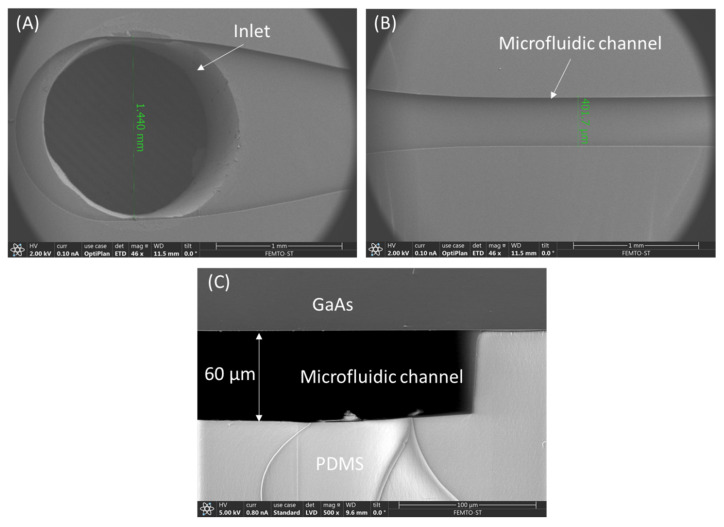
Characterization of microfluidic channels by SEM: (**A**) General view from the top side (scale 1 mm) on PDMS membrane with intlet/outlet holes, (**B**) microfluidic channel, and (**C**) cross section of GaAs/SiO_2_ bonded to PDMS (scale 100 µm).

**Figure 4 micromachines-13-00755-f004:**
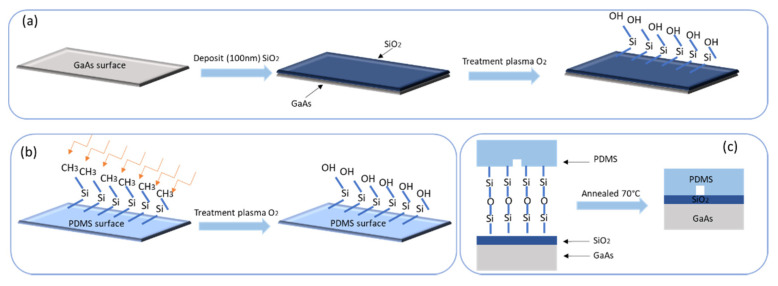
(**a**,**b**) Schematic presentation of GaAs/SiO_2_ and PDMS surfaces modifications by plasma O_2_; (**c**) bonding structure and annealing at 70 °C.

**Figure 5 micromachines-13-00755-f005:**
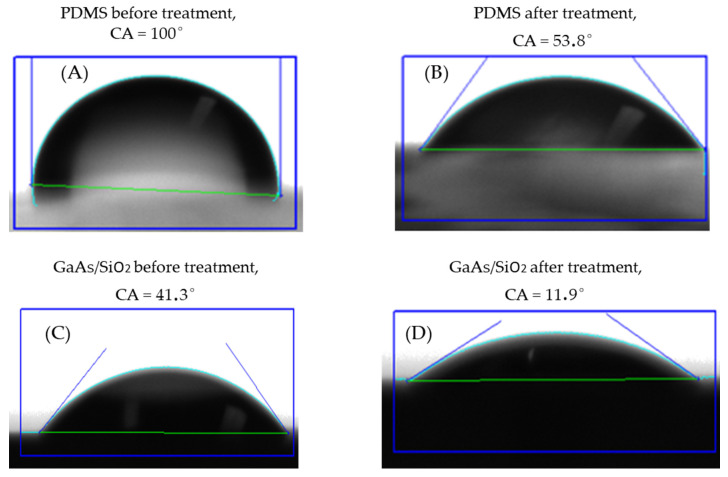
Contact angles for water droplet on the different surfaces; (**A**) PDMS before treatment; (**B**) PDMS activated by plasma treatment; (**C**) GaAs/SiO_2_ before plasma O_2_; (**D**) after plasma O_2_ treatment.

**Figure 6 micromachines-13-00755-f006:**
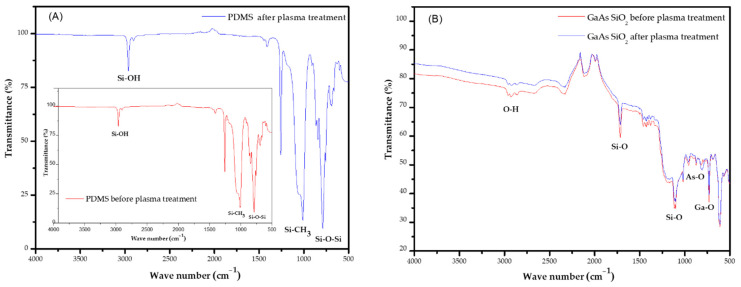
ATR-FTIR spectra recorded from: (**A**) PDMS polymer before and after plasma treatment; (**B**) the GaAs/SiO_2_ (100) surface before and after plasma treatment.

**Figure 7 micromachines-13-00755-f007:**
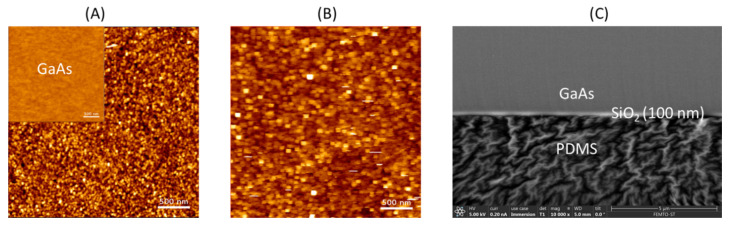
Surface morphology of SiO_2_ thin film deposited on GaAs substrates (**A**) before and (**B**) after O_2_ plasma treatment (AFM images, 3 × 3 μm^2^, contact mode, silicon nitride tips (0.32 N/m), 256 × 256 pixels resolution, scale 500 nm); (**C**) SEM image of bonding interface (scale 5 µm).

**Figure 8 micromachines-13-00755-f008:**
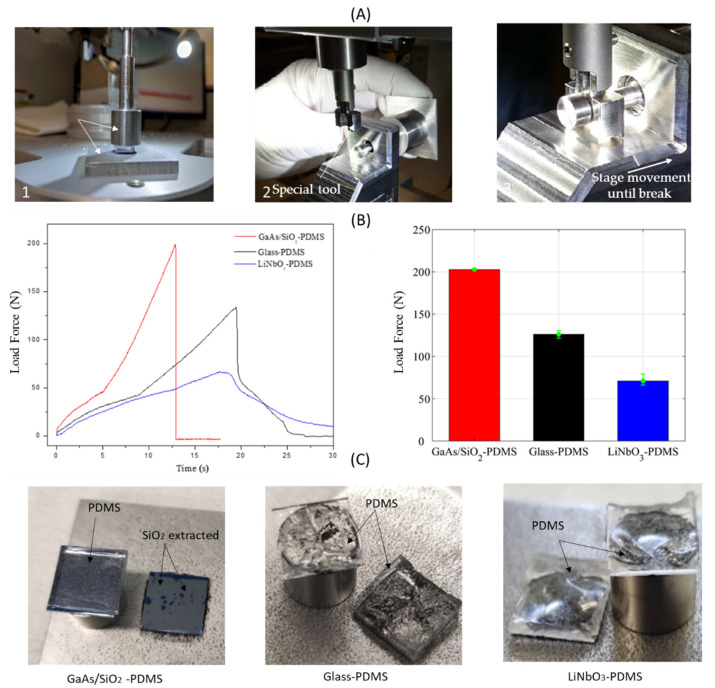
Bonding strength evaluation. (**A**) Setup of the tensile test: (1) assembly of the bonded sample: The sample is fixed on two steel sample holders, the die is placed inside, the cap rod is screwed into the block, and the die is glued between support plate and block; (2) position of the tool behind stud pin; (3) the sample mounting for the setup of the tensile test. (**B**-left) Load vs. time curves obtained for all substrates bonded to PDMS; (**B**-right) reproducibility histogram for bonding strength. (**C**) Photos of the broken interface for the PDMS-SiO_2_/GaAs, PDMS-Glass, and PDMS-LiNbO_3_ samples.

**Figure 9 micromachines-13-00755-f009:**
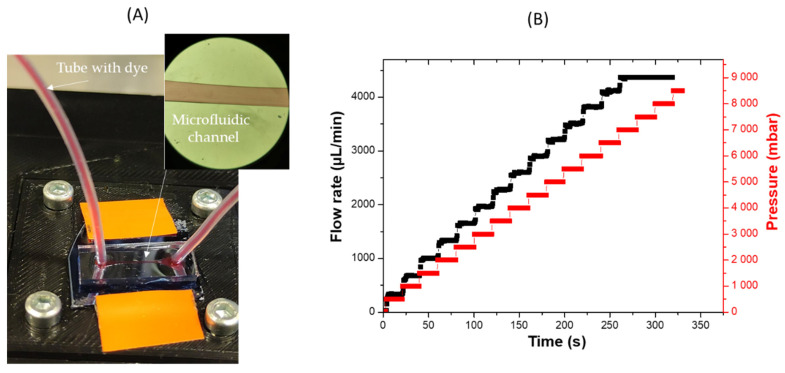
(**A**) Image of single microfluidic channel (height = 60 µm, width 300 µm, length 3 cm) infused with a red dye at 4500 µL/min and 8500 mbar pressure. (**B**) Increase in working pressure and flow rate on the microchannel.

**Table 1 micromachines-13-00755-t001:** The root mean square Average Roughness Rq and the average roughness Ra values for various substrates (PDMS value from literature [45]).

Sample	RMS Average Roughness Rq (pm)	Average Roughness Ra (pm)
GaAs/SiO_2_	359	552
GaAs	56	90
GaAs/SiO_2_ after plasma O_2_	293	441
PDMS	1441	not reported
PDMS treated by plasma O_2_	40,031	not reported

**Table 2 micromachines-13-00755-t002:** Leakage-free pressures values obtained from previously reported methods.

Sample	Pressure (Bar)	Method to Bon to PMDS	Ref
GaAs	≤8.5	plasma O_2_, SiO_2_, annealed	This work
Glass	5.1	plasma oxygen ICP	[50]
PDMS	6.7	plasma oxygen RIE	[49]
SU-8	1.5	plasma oxygen, small amount of PEIE and temperature	[38]
Glass/Au	2.38	plasma oxygen and narrow electrode	[28]
TPE	4.7	plasma oxygen and thermal bonding	[58]
PMMA	2.5	plasma oxygen	[55]

## Data Availability

Not applicable.

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
