# Peer review of "Microfluidics for High Pressure: Integration on GaAs Acoustic Biosensors with a Leakage-Free PDMS Based on Bonding Technology"

_micromachines, 2022, doi:10.3390/mi13050755_

Round 1

Reviewer 1 Report

The manuscript proposes an irreversible and leakage -free plasma -assisted bond to incorporate PDMS microfluidics on GaAs substrates using SiO2 thin films. The post -treatment process involves plasma O2 and low -temperature annealing. The topics discussed are novel and are believed to be beneficial to microfluidic technology industry. However, after reading the article carefully, we have concluded that the article has shown effort. However, it still needs some changes before it is ready to be published.

  1. First and foremost, the proposed bonding technology involves annealing at a low temperature of 70 °C. However, there is no optimization analysis on the annealing temperature. It is possible that the optimal annealing temperature is less than 70 °C. Please explain such an argument.
  2. In Figure 3, please clearly state the thickness or width of the microfluidic channel. In addition, the scale should be clear and proportionate.
  3. Please improve the resolution quality of Figures 5 and 6. Authors should adhere to the guidelines by providing Figures with at least 300 dpi.
  4. For statistically acceptable surface morphological analysis using AFM, authors should provide at least 5 different scan locations or places to confirm the RMS surface roughness. Since these results are among the main results, an acceptable statistical analysis is mandatory. In addition, please include the error of the measured RMS value.
  5. A similar statistical analysis is also required in the “bond strength evaluation” analysis. The author should provide at least 5 different measurements for each setup, and then the figures convinced by the accompanying errors can be plotted as in Figure 8 (b).6.
  6. There are few minor format errors, for example, values and units should be spaced: line 16, line 74, line 116 and etc. It is also proposed to improve the resolution quality of all images and redraw all the tables in the article according to scientific manuscript standards. Please revise the article accordingly.

Author Response

We would like to thank the reviewer for his critical comments. Accordingly, we have prepared a point-by-point response to the comments provided. Our responses are indicated in blue below the reviewer’s comments.

Reviewer 2 Report

Recommendation: Major revision

The authors reported a plasma-assisted method for bonding between PDMS microfluidic channel and GaAs/SiO2 substrate at low temperature. They evaluated the mechanical and fluidic performances of the fabricated devices and characterized the bonding surfaces by water contact angle measurement, ATR-FTIR, AFM and SEM analysis. In the end, the authors characterized the bonding strength by using a tensile machine and conducted a pressure/leakage test. Overall, the topic itself is interesting, and the work is quite thorough. However, the reviewer has some concerns with this work and the authors should address them before the paper can be accepted.

Major comments:

  • Innovation: The reviewer appreciates the authors’ effort for evaluating the PDMS bonding strength with different substrates (LiNbO3, glass, and GaAs/SiO2). However, the deposition of SiO2 as an intermediate layer to enhance the bonding strength between PDMS and piezoelectric substrate (Two-dimensional single-cell patterning with one cell per well driven by surface acoustic waves. Nature communications, 6(1), 1-11, 2015) is not new. The innovation of PDMS bonding with GaAs/SiO2 substrate should be better elucidated in the introduction.
  • Significance: By depositing SiO2 intermediate layer, many parameters, such as acoustic velocity, CTE, Young's modulus, surface roughness, etc., may change. And some of these changes can have significant influence on the performance of the fabricated acoustic sensors or actuators. Why is the deposition of SiO2 intermediate layer better than other methods, especially the APTES one? Since the APTES method has minimum influence on the performance with no structural changes of the substrate. In addition to PMMA/PDMS bonding, APTES (Line 99) has been used to increase the bonding strength between PDMS and piezoelectric substrate (Harmonic acoustics for dynamic and selective particle manipulation. Nature Materials, (2022), https://doi.org/10.1038/s41563-022-01210-8), and the authors should discuss the corresponding method.

Detailed comments:

  • Title: The claim of “GaAs acoustic biosensors” is misleading. There is very little introduction/discussion regarding the design, fabrication, or characterization of any acoustic biosensors.
  • Line 20: The use of commas in many numerical values (2,06 MPa-->2.06 MPa? (Line 20) and 2,950 cm−1 ? (Line 228)) is confusing.
  • Line 131: The authors should double check the description "Diamond saw dicing was used to cut PDMS" (Si mold?). Blade cutting is typically used for PDMS, as shown in Line 181.
  • Line 158: Why sputtering instead of e-beam evaporation or PECVD is used for SiO2 deposition? The adhesion of SiO2 layer on GaAs substrate seems not good (Fig. 8c).
  • Figure 3: Figures with higher resolution should be provided. The reviewer cannot distinguish the labels in green color. Fig. 3c: SiO2 intermediate layer should be indicated.
  • Figure 5: Scale bar should be provided.
  • Figure 6a: The reviewer cannot distinguish the difference between the blue and red curves. A zoomed inset figure will be helpful. (A) and (B) should be labeled in this figure.
  • Figure 7c: The reviewer cannot distinguish the labels in green color.
  • Figure 8: Why PDMS bonded on the GaAs device was glued in a square sample holder, while the other two are not? The label (A) is missing.

Author Response

(The authors gave the same response as above.)

Round 2

Reviewer 2 Report

The authors have addressed the reviewer's comments.